# Novel *Bacillus*-Infecting Bacteriophage B13—The Founding Member of the Proposed New Genus *Bunatrivirus*

**DOI:** 10.3390/v14102300

**Published:** 2022-10-19

**Authors:** Olesya A. Kazantseva, Emma G. Piligrimova, Andrey M. Shadrin

**Affiliations:** Laboratory of Bacteriophage Biology, G. K. Skryabin Institute of Biochemistry and Physiology of Microorganisms, Pushchino Scientific Center for Biological Research of the Russian Academy of Sciences, Federal Research Center, 142290 Pushchino, Russia

**Keywords:** bacteriophage, phage, *Caudoviricetes*, *Bacillus*, *Bacillus cereus sensu stricto*, *Bacillus cereus sensu lato*

## Abstract

In this work, we describe a novel temperate bacteriophage, *Bacillus* phage B13. *Bacillus*-infecting phages are widespread and abundant, though often overlooked including because of their temperate lifestyle. B13 was isolated from its bacterial host via mitomycin C induction. Its host range was determined, and its pH and thermal stability were evaluated. The whole genome of B13 was sequenced and annotated. The genome is 36,864 bp long and contains 53 genes. The tail genes of B13 suggest that the phage has a siphovirus morphotype. It was found both in vitro and in silico that the phage uses the 3′-cos DNA packaging strategy, and the phage genome termini were located. Comparative analyses revealed that B13 has no close relatives and should therefore be assigned to a new viral genus, for which we propose the name *Bunatrivirus*.

## 1. Introduction

Bacterial viruses (bacteriophages) are the most abundant biological entities on Earth [1]. As of August 2022, the class *Caudoviricetes* (tailed bacteriophages: realm *Duplodnaviria*, kingdom *Heunggongvirae*, phylum *Uroviricota*, class *Caudoviricetes*) encompasses 1197 phage genera and 3601 phage species (the International Committee on Taxonomy of Viruses web site: http://ictv.global (accessed on 1 August 2022)) [2]. New phages are being discovered at an increasing pace, and the number of phage taxa established by the International Committee on Taxonomy of Viruses (ICTV) is growing by the year [3,4]. Despite the considerable amount of knowledge that the scientific community has already accumulated on the diversity of bacteriophages, experts agree that the majority of phage taxa remain to be created [5,6,7].

Exceptionally ubiquitous and notably diverse are bacteriophages preying on *Bacillus*. *Bacillus* is a genus of Gram-positive, rod-shaped, spore-forming bacteria inhabiting various ecological niches. Many *Bacillus* species have found their uses in different fields; *B. thuringiensis* is widely used in agriculture due to its insecticidal properties [8], and at least seven *Bacillus* species are used as human probiotics [9] and food supplements in livestock husbandry [10]. However, some representatives of the genus are no friends of ours, such as *Bacillus anthracis*, a risk group 3 human pathogen and the causative agent of anthrax. Conditionally pathogenic *Bacillus cereus sensu stricto* is another potential enemy; being an omnipresent food contaminant able to grow at refrigerator temperatures, it is responsible for up to 12% of all food poisoning outbreaks worldwide [11].

Tailed *Bacillus*-infecting bacteriophages have different morphological features, genome sizes, and lifecycle strategies [12]. They are well-represented in the official virus taxonomy and, along with phages infecting *Enterobacteriales*, comprise many genera of the class *Caudoviricetes*. However, the already described and taxonomically classified phages of *Bacillus* seem to be only a small fraction of the existing diversity, as reports keep appearing on newly discovered *Bacillus*-infecting phages with very few similarities to any of the known viruses [13,14,15,16].

From the taxonomical viewpoint, temperate phages may be particularly interesting, which, in the prophage form, are easy to overlook when relying on conventional cultivation-based approaches for phage identification. Prophages can contribute to the viability and adaptability of their bacterial host by altering some of its major characteristics such as antibiotic resistance, metabolic pathways, pathogenicity, and anti-phage resistance [17,18]. With the average *Bacillus* genome containing around five prophages [18], it is conceivable that an extensive exploration of prophage sequences paired with some experimental research to prove the prophages’ viability would very soon enable us to significantly increase the number of taxa comprising *Bacillus*-infecting viruses.

In the present study, we describe a novel temperate *Bacillus*-infecting bacteriophage, B13. The phage was isolated from its bacterial host (*Bacillus cereus* VKM B-13) through mitomycin C induction, and its whole-genome sequence was determined. The phage is too genetically remote from the known viruses to be classified into any of the existing phage genera. Therefore, we propose creating a new genus, *Bunatrivirus*, to formally classify B13.

## 2. Materials and Methods

### 2.1. Bacterial Strains

The bacterial strains used in this study (Appendix A) were acquired from the All-Russian Collection of Microorganisms (VKM). The strains and the phage were cultivated in Lysogeny broth (LB) (10.0 g/L tryptone; 5.0 g/L yeast extract; 10.0 g/L sodium chloride) and on LB agar (1.5% *w/v* and 0.5% *w/v*) (10.0 g/L tryptone; 5.0 g/L yeast extract; 10.0 g/L sodium chloride; 15.0 g/L bacto agar or 5.0 g/L bacto agar, respectively) with the addition of 10 mM CaCl_2_ and 10 mM MgCl_2_. All cultures were grown at 28 °C with shaking at 120 rpm.

### 2.2. Phage Isolation and Propagation

The induction of phage B13 from the lysogenic strain *B. cereus* VKM B-13 was performed using mitomycin C. Briefly, the bacterial strain was grown in a well of a 48-well microplate, in 495 µL of LB broth with 10 mM CaCl_2_ and 10 mM MgCl_2,_ until the optical density reached 0.2 at 595 nm. The microplate was incubated with shaking at 28 °C in a FilterMax F5 microplate reader (Molecular Devices), with OD_595_ measured every 10 min. Then, 5 µL of mitomycin C (50 μg/mL) was added to a final concentration of 0.5 μg/mL. The culture was incubated at 28 °C until optical density decreased. The cell debris was removed by centrifugation at 3000 g for 12 min at 4 °C, and 3 μL of chloroform was added to the resulting supernatant. The supernatant was titrated with serial dilutions on the lawn of the sensitive strain *B. cereus* VKM B-370 using the double agar layer technique, and the plate was incubated at 28 °C overnight. Separate plaques were mixed with 750 μL of SM+ buffer (50 mM Tris-HCl, pH 8.0; 100 mM NaCl; 1 mM MgSO_4_; 0.1% gelatin; 10 mM CaCl_2_; 10 mM MgCl_2_) and 75 μL of chloroform in a 1.5-mL Eppendorf tube followed by incubation overnight at 4 °C for phage extraction. The obtained phage extracts were briefly centrifuged at 3000 g. In order to propagate the phage, 50 µL of each phage extract was mixed with 5 µL of a *B. cereus* VKM B-370 culture (OD = 1.2) and 445 µL of LB broth with 10 mM CaCl_2_ and 10 mM MgCl_2_ in a well of a 48-well microplate. A non-infected bacterial culture (with 50 µL of SM+ buffer added instead of a phage extract) was used as a control to assess the phage activity. The microplate was incubated under the same conditions mentioned above. After that, the lysates were centrifuged at 3000 g for 10 min at 4 °C, and 10 μL of chloroform was added to the supernatants. The obtained phage suspensions were used in a plaque assay, which was performed using the double-layer agar technique. Plates were incubated overnight and examined for plaques. Separate plaques were selected and used for further phage propagation. The extraction–propagation cycle was repeated three times so as to make sure the phage suspension was not contaminated with other bacteriophages.

The strain *B. cereus* VKM B-370 (GenBank Accession Number: NZ_CP070339.1) was used to propagate the phage in order to obtain a higher titer preparation. Briefly, 100 μL of a VKM B-370 culture (OD = 1.2) was transferred into a 250-mL flask with 50 mL of LB with 10 mM CaCl_2_ and 10 mM MgCl_2_ and incubated at 28 °C with shaking at 120 rpm until an optical density of 0.1 was reached. Then, 50 μL of the obtained phage suspension was added. The phage–bacterial mixture was incubated for 6 h with periodic monitoring of optical density. After that, 2.8 g of NaCl and 500 μL of chloroform were added, and for 30 more minutes, the incubation continued. The mixture was then centrifuged at 10,000× *g* for 10 min to remove the cell debris, followed by adding polyethylene glycol (PEG) 8000 to the final concentration of 10% for phage precipitation. After an overnight incubation at 4 °C, the mixture was again centrifuged at 10,000× *g* for 10 min, and the precipitate was resuspended in 5 mL of SM+ buffer and filtered through a 0.22 µm filter. The obtained PEG-concentrated preparation was stored at 4 °C.

### 2.3. Host Range Test

Phage host specificity was evaluated on 38 strains of the *B. cereus* group. For this purpose, 30 μL of the frozen stock bacterial cultures in the late exponential growth phase was mixed with 3 mL of molten soft LB agar (0.5% *w/v*) with 10 mM CaCl_2_ and 10 mM MgCl_2_, then vortexed and overlaid on solid LB agar (1.5% *w/v*) in 6-cm Petri dishes. Next, 3 μL of a B13 suspension (10^6^ plaque-forming units (PFU)/mL) was dripped onto the agar plates with the host strains, and the plates were incubated for 24 h at 28 °C.

### 2.4. Killing Assay

To assess the killing activity of phage B13, exponential-phase cultures of the sensitive strain *B. cereus* VKM B-370 were infected with the phage at different MOI. Briefly, 50 μL of four B13 preparations (with phage titer of 2 × 10^8^, 2 × 10^7^, 2 × 10^5^, and 2 × 10^5^ PFU/mL) was added separately to 450 μL of mid-log bacterial cultures (at approximately 2 × 10^6^ colony-forming units (CFU)/mL) to ensure a multiplicity of infection (MOI) of 10, 1, 0.1, and 0.01, respectively. The mixtures were incubated in a 48-well microplate with shaking at 28 °C for 10 h in a FilterMax F5 microplate reader (Molecular Devices), while measuring OD_595_ every 10 min. Non-infected *B. cereus* VKM B-370 cultures were used as controls to assess the killing activity. Three independent trials of the experiment were performed. The results were presented as the mean of three observations ± standard deviation, and visualized in the form of growth curves using GraphPad Prism 8.4.3.

### 2.5. Thermal and pH Stability Tests

Phage stability assessments were carried out as described previously [16], with some modifications. In order to assess the stability of phage B13 at different temperatures, aliquots of a phage preparation (at approximately 3 × 10^7^ PFU/mL) were incubated at 4, 30, 40, 50, 60, 70, 80, and 90 °C for 1 h, and the surviving phages were enumerated by a spot test. Phage stability under different pH values (ranging from 2.2 to 12) was evaluated using five buffers: Glycine-HCl buffer (pH values 2.2 and 3), sodium acetate buffer (pH 4 and 5), phosphate buffer (pH 6, 7 and 8), Glycine-NaOH buffer (pH 9–10), and Na_2_HPO_4_-NaOH buffer (pH 11–12). Phage preparation mixed with SM+ buffer was used as a control. Aliquots of a B13 preparation were added to the buffers (final phage concentration of 3 × 10^7^ PFU/mL) and incubated at 30 °C for one hour. The number of viable plaque-forming units in each solution was quantified by spot test. Three independent trials of the experiments on thermal and pH stability were carried out. The results were processed in GraphPad Prism 8.4.3 and presented as the mean of three observations ± standard deviation.

### 2.6. Genome Sequencing, Assembly and Sequence Analysis

A B13 phage preparation (6 × 10^8^ PFU/mL) was incubated with DNase I and RNase A according to the manufacturer’s protocol. Phage DNA was extracted using the DNeasy Blood & Tissue Kit (Qiagen) according to the manufacturer’s instructions. A sequencing library was prepared using the TruSeq DNA Library Preparation Kit (Illumina Inc, San Diego, CA, USA) and sequenced on the Illumina MiSeq platform (Illumina Inc, San Diego, CA, USA). Sequencing reads were trimmed using Trimmomatic v0.39 [19]. For genome assembly, SPAdes v. 3.12.0 was used [20]. Open reading frames (ORFs) were identified with RASTtk v.2.0 [21]. The putative functions of ORFs were predicted using BLASTp (NCBI) [22] and HHpred [23]. ARAGORN v1.2.41 [24] was used for tRNA and transfer-messenger RNAs gene identification. The B13 genome map was visualized with BRIG v. 0.95 [25]. Phage protein homology was analyzed using BLASTp (NCBI) [22].

### 2.7. The Genome Packaging Strategy

In order to identify the genome packaging strategy of phage B13, a phylogenetic analysis of the phage’s large terminase subunit was performed as described by Merrill [26]. For this purpose, 58 large terminase amino acid sequences were aligned using MAFFT v7.490 [27] with default settings. A phylogram was inferred from the alignment in MEGA X [28] using the Neighbor-Joining method with the bootstrapping parameter set to 1000. The large terminase subunit phylogram was visualized in FigTree v1.4.4 [1] with midpoint rooting.

The type of genome termini was predicted computationally using the ‘High Occurrence Read Termini’-based approach [29,30] implemented in the PhageTerm tool [31]. The predicted termini were experimentally validated in a standard restriction analysis [32].

### 2.8. Comparative Genomics

To find related phages, a ViPTree analysis was conducted based on genome-wide sequence similarities computed by tBLASTx [33], and a viral proteomic tree was generated. As the ViPTree database lacks some of the phage genomes available in the GenBank database, six phage genomes found in a BLASTn search against the whole-genome sequence of B13 were added to the VipTree analysis (Appendix A). A linear diagram illustrating the level of identity between the B13 genome and the most closely related genomes was also created using the ViPTree server version 3.1 [33]. The percentage of shared proteins was computed using the GET_HOMOLOGUES software v3.3.3 [34] with the COGtriangles algorithm [35] with a threshold of 75% for query coverage and 1 × 10^−5^ for E-value on the all-against-all BLASTp results (-t 0 –C 75 -e).

### 2.9. Accession Number

The genome sequence of phage B13 was deposited into the GenBank database under accession number OP066531 (BioProject accession number PRJNA861678, BioSample accession number SAMN29921977). The raw sequencing reads can be found in the Sequence Read Archive using accession number SRR20755294.

### 2.10. Statistical Analysis

Experimental results were statistically evaluated in order to ascertain their statistical significance. Statistical analyses were performed using GraphPad Prism 8.4.3. Two-way ANOVA with repeated measures was applied to analyze the differences in bacterial growth rates upon B13 infections at different MOI (3.2. subsection, *Killing Assay*). One-way ANOVA with repeated measures was used to evaluate the significance of the observed differences in phage concentrations in the phage stability experiments (3.3. subsection, *pH and Thermal Stability Tests*). Statistically significant results were those with *p* ≤ 0.05.

## 3. Results and Discussion

### 3.1. Phage Isolation and Host Range

*Bacillus* phage B13 was isolated from its host strain *B. cereus* VKM B-13 by mitomycin C prophage induction. To determine the host range of B13, the purified phage preparation with a titer of 10^6^ PFU/mL was tested on 38 strains of the *B. cereus* group. Twenty-one out of the thirty-eight strains (55.3%) were sensitive to the phage (Appendix A). On the propagating host *Bacillus cereus* VKM B-370 (GenBank Accession number: NZ_CP070339.1), B13 formed turbid plaques around 1–1.5 mm in diameter (Figure 1), suggesting a temperate lifestyle.

### 3.2. Killing Assay

The growth kinetics of *B. cereus* VKM B-370 upon infection with B13 at different MOI was studied by comparing the optical density (OD_595_) values of phage-infected bacterial cultures with those of non-infected control cultures. The resultant growth curves are shown in Figure 2. The higher the MOI, the faster partial growth inhibition was reached. Even at the highest MOI (10), the phage did not ensure complete lysis of the culture; after temporary inhibition, bacterial growth resumed and continued until reaching a plateau (Figure 2). The optical densities of all the infected and control bacterial cultures leveled off after about 9 h of cultivation (Figure 2). Such lysis dynamics—no sustainable growth inhibition—is indicative of temperate phages.

### 3.3. pH and Thermal Stability Tests

The phage B13 was highly stable during one-hour incubation in an alkaline environment as opposed to an acidic one (Figure 3a). No significant change in the phage titer was detected in solutions ranging from pH 6 to pH 11 (in comparison with a control sample, SM+) (Figure 3a), while no phages survived incubation under acidic conditions at pH 2.2, 3, 4, and 5 (Figure 3a) (ANOVA, *p* < 0.05).

The phage survived at temperatures of 30, 40, and 50 °C. The phage titer was rather stable during one-hour incubation at temperatures of 30 and 40 °C, while incubation at 50 °C led to about an order of magnitude decrease in the titer, compared to the control sample incubated at 4 °C (ANOVA, *p* > 0.05) (Figure 3b). No phages survived temperatures of 60 °C and higher, as no plaques were observed on the bacterial lawn.

### 3.4. General Genome Organization

Phage DNA was sequenced on the Illumina platform, and the genome was assembled using SPAdes v. 3.12.0 [20]. The B13 chromosome is a linear dsDNA molecule with a length of 36,864 bp and a GC content of 34.8%. The genome contains 53 CDSs, as predicted by RASTtk v.2.0 [21]. Thirty out of the fifty-three CDSs (56.6%) were functionally annotated using BLASTp and HHpred and classified into the following functional modules: DNA packaging and structural module, lytic module, DNA replication and recombination module, prophage maintenance module, and transcription regulation module (Figure 4). For convenient visualization, the genome map is shown in Figure 4 as a circular diagram with the first base of the small terminase subunit gene placed at the beginning of the genome.

The whole-genome arrangement of B13 is typical of temperate phages, with the majority of genes (48 genes; 90.6%) located on one strand and five on the opposite strand. Among the latter is a site-specific tyrosine integrase gene surrounded by transcriptional regulator genes.

The whole-genome sequence of the host strain *Bacillus cereus* VKM B-13 was obtained (unpublished data), and the whole genome of phage B13 was localized in one of the sequencing contigs, surrounded by the host DNA sequences—an evidence of the integrated state of the B13 prophage. The phage integration site was identified: it lies 44 bp downstream of the integrase gene in the B13 genome and has the following sequence: 5′- AATCGTTCGTTCTATAA -3′.

#### 3.4.1. DNA Packaging Genes and Genome Packaging Strategy

The DNA packaging gene module includes two genes: one coding for a small terminase subunit (CDS1; protein_id UUW40187), a protein which binds to the packaging initiation site and regulates the ATPase activity of a large terminase subunit; and the other encoding a large terminase subunit (CDS2; protein_id UUW40188), the main packaging protein of tailed bacteriophages, which is responsible for the ATP-powered translocation of phage genomic DNA into phage procapsids. Large terminase subunit comparisons are used in phage taxonomic classification and can also be applied to predict phage DNA packaging strategies and types of genome ends. In order to predict the packaging mechanism of B13, the large terminase protein of B13 was compared with those of phages with well-studied DNA packaging mechanisms [32] (Appendix A). As shown in Figure 5, the large terminase protein of B13 shares one clade with those of phages *Escherichia* virus HK97, *Escherichia* virus HK022, and *Rhizobium* phage 16-3, which are known to use the 3′-cos packaging mechanism.

The genome termini and DNA packaging mechanism of B13 were also predicted bioinformatically using PhageTerm [31]. According to the PhageTerm analysis, the B13 terminase can recognize a specific site on the phage DNA and generate fixed DNA termini with 3′ overhangs with the following sequence: 5′-GGACCGAGAGGGGA-3′. Thus, according to PhageTerm, B13 uses the 3′-cos packaging mechanism similarly to the well-known *Escherichia* phage HK97. The prediction was confirmed experimentally through a DNA restriction analysis: phage DNA was digested with HindIII, BamHI, BgIII and PstI *in silico* (Figure 6a) and *in vitro* (Figure 6b), and the resultant restriction patterns were compared. The two patterns were identical except for the case with HindIII digestion, where an additional band appeared *in vitro* with a length of 3.7 Kb (indicated with red asterisk in Figure 6b). This band could be formed by the left (3143 bp) and right (552 bp) end fragments (indicated by green and blue arrows, respectively, in Figure 6b) annealed together by their cohesive ends during the gradual temperature decrease after the HindIII restriction reaction. In the case of BamHI, BgIII, and PstI, the annealed left and right end fragments were too difficult to detect due to the resolution limitations of the method. The observed type of restriction pattern is indicative of phages using the 3′-cos mode of DNA packaging [32,36].

Thus, based on the results of phylogenetic, bioinformatic, and restriction analyses, it can be concluded that B13 uses an HK97-like 3′-cos mechanism of DNA packaging with the cohesive end sequence being 5′-GGACCGAGAGGGGA-3′.

#### 3.4.2. Structural, Morphogenesis and Lytic Genes

The identified structural proteins of B13 and proteins involved in the phage’s capsid assembly include: portal protein (CDS3, UUW40189), procapsid protease (CDS4, UUW40190), major capsid protein (CDS5, UUW40191), and two head terminator proteins (CDS7, UUW40193 and CDS8, UUW40194). Proteins comprising the phage’s tail and those involved in its assembly include: tail terminator protein (CDS10, UUW40196), tail tube protein (SDC11, UUW40197.1), tail assembly chaperone (CDS12, UUW40198), tape measure protein (CDS14, UUW40200), distal tail protein (SDC15, UUW40201), and tail fiber protein (CDS16, UUW40202). The tail genes of B13 are typical of the siphovirus morhotype: bacteriophages with a long, non-contractile flexible tail [37].

The tail fiber proteins of B13 (Gp16) contains a predicted N-terminal endopeptidase domain. Endopeptidase domains are often found in the tail fiber proteins of bacteriophages infecting Gram-positive bacteria, where they serve to enable phage DNA entry into the cell; they cleave amide bonds within the cross-linked peptides connecting two or more glycan chains, which results in holes in the peptidoglycan layer. Phage tail hydrolases, along with phage endolysins destroying the bacterial cell wall from within, can be used to fight bacterial pathogens. According to a BLASTp analysis, the B13 tail fiber protein shares low similarity with those of the *Bacillus* phages phi4B1 (YP_009206320.1; 39.24%), phi4J1 (YP_009218150.1; 68.89%), and vB_BtS_B83 (YP_009845458.1; 55.09%). The lytic properties of these proteins have not yet been investigated, and it may be the objective of one of our further studies to evaluate the lytic capabilities of B13 Gp16.

Lytic proteins encoded in the phage genome include: holin (CDS18, UUW40204) and endolysin of the N-acetylmuramoyl-L-alanine amidase type (CDS19, UUW40205). Phage lytic proteins play an important role in the phage lifecycle since they are involved in the bacterial cell wall destruction at the stage of phage progeny release. These proteins can be used as potential agents to inhibit or eliminate bacterial pathogens. The B13 holin shares significant similarity with those of the following *Bacillus* phages: phi4J1 (YP_009218152.1; 89.36% aa identity), 250 (YP_009219599.1; 88.65%), BVE2 (AUG88607.1; 87.94%), IEBH (YP_002154392.1; 88.65%), vB_BtS_BMBtp13 (ANT39958.1; 87.94%), AP631 (AZF88361.1; 86.52%), phIS3501 (YP_007004376.1; 86.52%), Gamma (YP_338199.1; 86.52%), and PfEFR-5 (YP_009285267.1; 77.25%). A BLASTp analysis of the N-acetylmuramoyl-L-alanine amidase of B13 revealed several closely related endolysins, including those of the *Bacillus* phages Waukesha92 (YP_009099314.1; 86.50% aa identity), vB_BthS-HD29phi (QDP43493.1; 73.51%), phiCM3 (YP_009009166.1; 68.48%), phi4J1 (YP_009218153.1; 53.86%), vB_BanS-Tsamsa (YP_008873459.1; 52.5%), Izhevsk (QIW89878.1; 52.5%), pW2 (AZU98917.1; 51.42%), and PBC2 (AKQ08512.1; 50.48%). The endolysin of phage Izhevsk, Ply57, was described in one of our previous studies [38], where we showed that it was twice as thermostable as PlyG from phage Wβ, one of the well-studied endolysins from phages infecting the *B. cereus* group of bacteria. Experimental investigation of the B13 endolysin, including its comparative analysis with endolysins from other *Bacillus* phages, such as Ply57, is of considerable interest and may be one of the directions of future research in our laboratory.

#### 3.4.3. Replication Genes

The B13 proteins responsible for replication are replication terminator protein (CDS35, UUW40221) and DnaD-like helicase loader (CDS38, UUW40224).

### 3.5. Comparative Genomics

In order to identify the phylogenetic relationship between B13 and the known viruses, a viral proteomic tree was constructed using the ViPTree server version 3.1. As can be seen in the tree (Figure 7), the closest relatives of B13 are the *Bacillus* phages BMBtp1 and vB_BanS_Athena.

As of 25 August 2022, no virus genome available in the GenBank database (taxid: 10239) demonstrated substantial similarity to B13. The closest BLASTn nucleotide identity of the *Bacillus*-infecting phage phi4J1 was found to be only 31.28% (calculated as % identity multiplied by % coverage). Other relatives are the *Bacillus* phages BMBtp1 and vB_BanS_Athena, which are 27.73% and 21.54% identical to B13, respectively. A tBLASTx pairwise comparison revealed that the regions of noticeable similarity (identity value of 40% or more) between the B13 and BMBtp1 genomes contain genes coding for prophage maintenance proteins, replication proteins, and transcription regulators, as well as genes with unknown functions, while the similarity between the B13 and vB_BanS_Athena genomes was observed in the region of genes coding for DNA packaging protein and structural proteins (Figure 8). BMBtp1 is the phage that shares the highest number (26) and percentage (46.85%) of proteins with B13, as was calculated using GET_HOMOLOGUES (Appendix A).

With the ICTV-established cut-off for genus demarcation being 70% total nucleotide identity [4], B13 clearly cannot be classified into any of the existing phage genera. Therefore, we propose the creation of a new phage genus, *Bunatrivirus,* to formally classify B13 within the framework of contemporary virus taxonomy.

## 4. Conclusions

In this study, the temperate *Bacillus*-infecting siphovirus B13 was isolated from its host, *Bacillus cereus* VKM B-13 through mitomycin C prophage induction. It was shown that the phage uses the 3′-cos mechanism of DNA packaging, and the sequence of the terminal overhangs was predicted. The phage cannot be assigned to any of the existing phage genera; the closest relative, *Bacillus* phage phi4J1, is only 31% identical to B13 on the nucleotide level. Therefore, in order for B13 to become part of the official virus taxonomy, we propose creating a new genus, *Bunatrivirus*, for the time being, comprising only one species: *Bunatrivirus* B13.

## Figures and Tables

**Figure 1 viruses-14-02300-f001:**
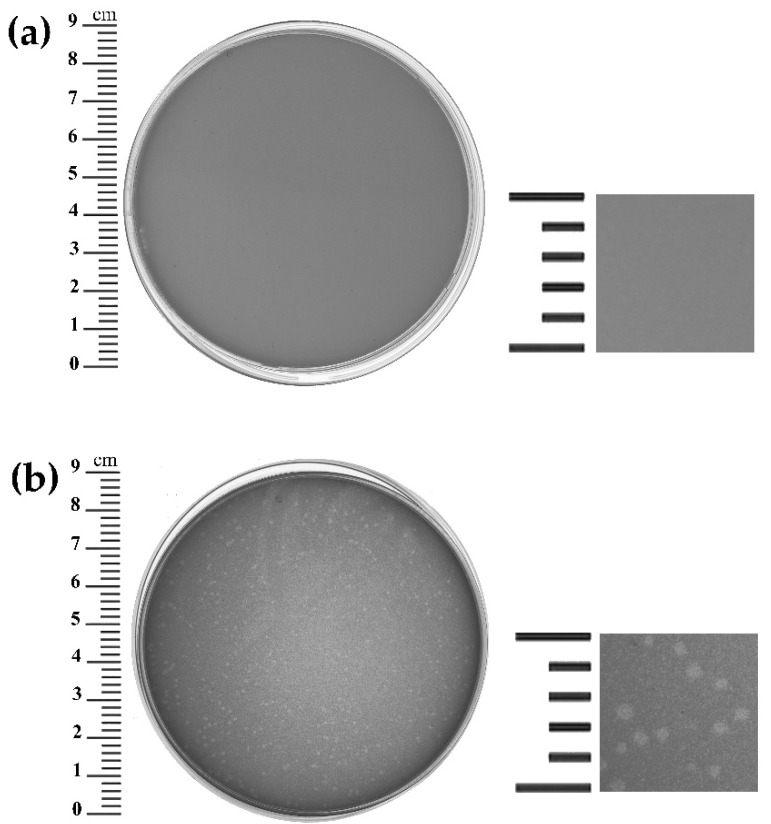
B13 plaque morphology on the lawn of *B. cereus* VKM B-370: (**a**) Control plate (no phage added); (**b**) Plaques formed by the B13 phage after 24-h incubation.

**Figure 2 viruses-14-02300-f002:**
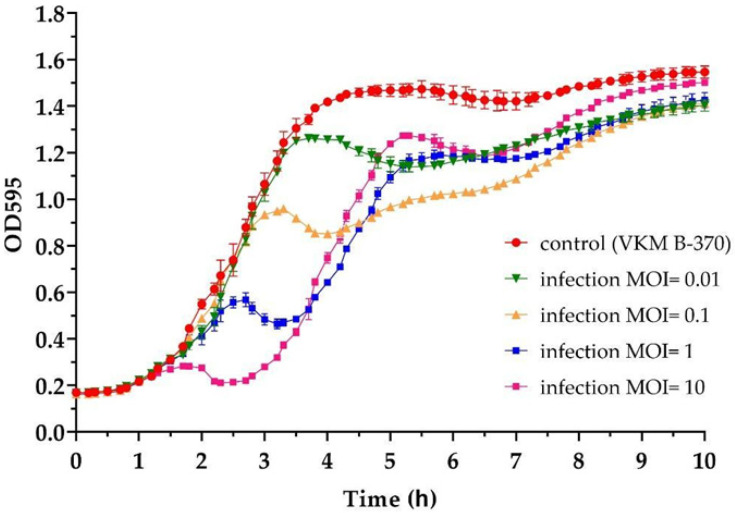
The growth curves of *B. cereus* VKM B-370 upon infection with B13 at different MOI values (see the legend). The graph was visualized in GraphPad Prism 8.4.3. Individual dots represent the mean of three independent trials, and the error bars represent the standard deviation.

**Figure 3 viruses-14-02300-f003:**
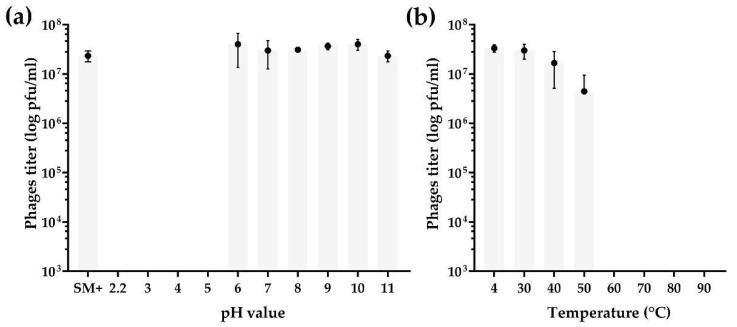
Phage stability tests of *Bacillus* phage B13: (**a**) pH stability and (**b**) thermal stability. The graphs were visualized in GraphPad Prism 8.4.3. Individual dots represent the mean of three independent trials, and the error bars represent the standard deviation.

**Figure 4 viruses-14-02300-f004:**
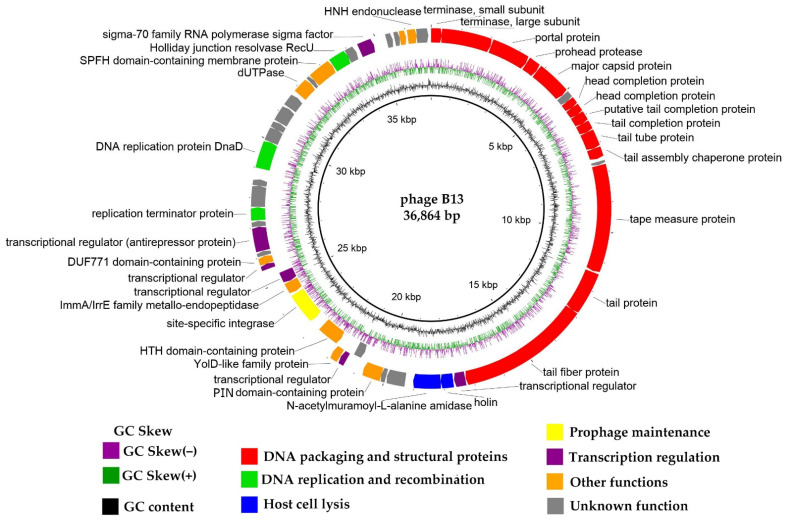
The *Bacillus* phage B13 genome map. Functionally assigned CDSs are highlighted based on their general functions (see the legend).

**Figure 5 viruses-14-02300-f005:**
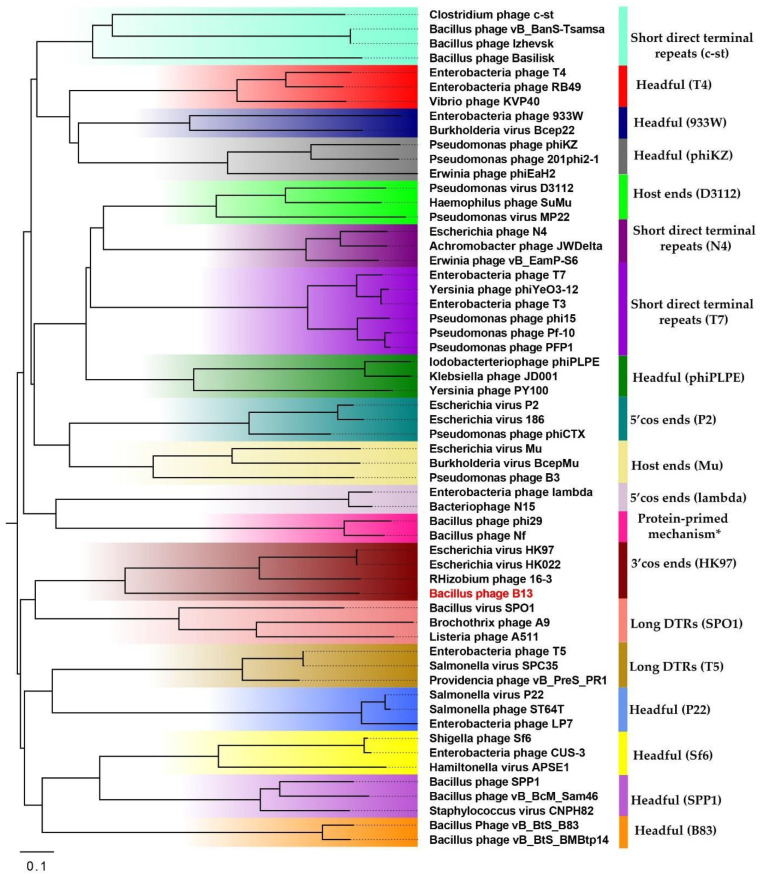
The phylogenetic analysis of the large terminase subunits of phage B13 and phages with well-studied packaging mechanisms. The tree was inferred in MEGA X using the Neighbor-Joining method with 1000 bootstrap replicates and visualized using FigTree v1.4.4.

**Figure 6 viruses-14-02300-f006:**
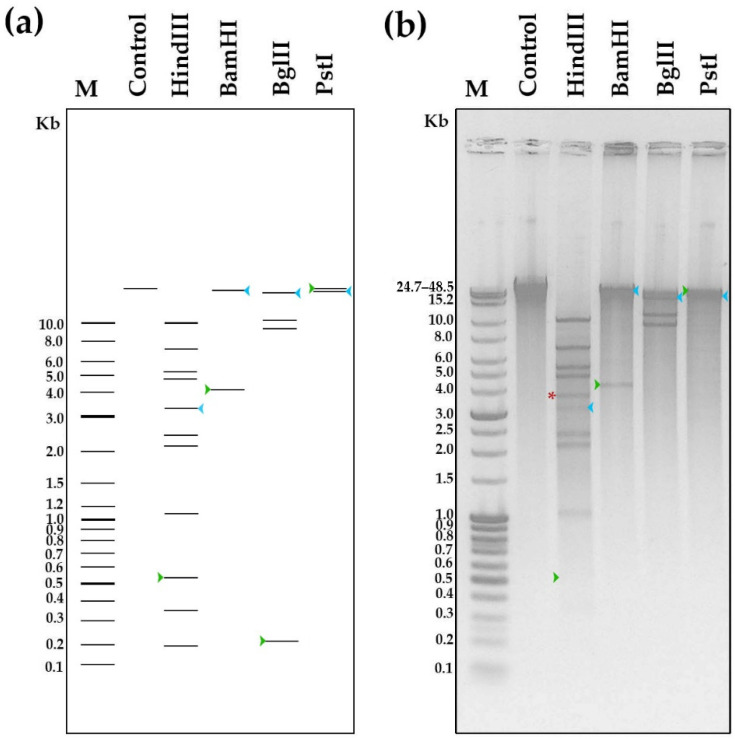
Phage DNA restriction analysis: (**a**) in silico and (**b**) in vitro. In silico digestion was performed using NEBcutter V2.0. M—molecular weight markers; control—intact phage DNA. Green and blue arrows indicate DNA fragments containing the B13 chromosome termini (left and right, respectively). A fragment formed by left and right terminal fragments annealed together by their cohesive ends is indicated by red asterisk.

**Figure 7 viruses-14-02300-f007:**
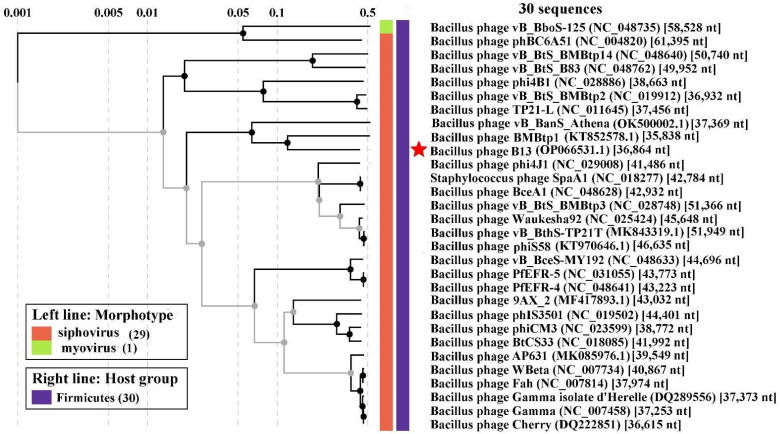
The viral proteomic tree including the *Bacillus* phage B13 and its closest relatives. The tree was inferred using ViPTree 3.1. The red star indicates the *Bacillus* phage B13.

**Figure 8 viruses-14-02300-f008:**
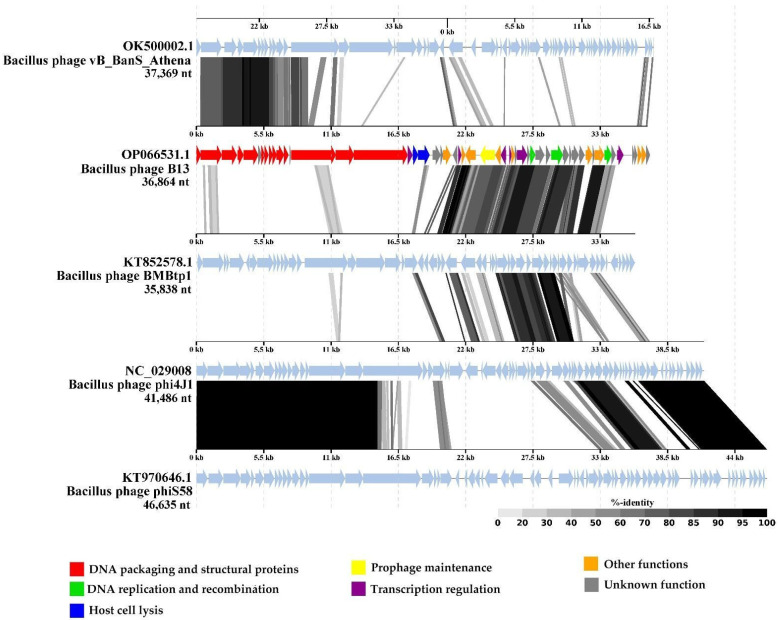
The pairwise tBLASTx comparison of the whole-genome sequences of B13 and its closest relatives visualized using the ViPTree server version 3.1. The B13 CDSs are colored based on their predicted functions (see the legend). Gray areas between the genome maps indicate the level of identity (0 to 100%, see the legend on the right).

## Data Availability

The genome sequence of phage B13 was deposited into the GenBank database under accession number OP066531 (BioProject accession number PRJNA861678, Bi-oSample accession number SAMN29921977). The raw sequencing reads can be found in the Sequence Read Archive using accession number SRR20755294.

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
