# Peer review of "Novel Bacillus-Infecting Bacteriophage B13—The Founding Member of the Proposed New Genus Bunatrivirus"

_viruses, 2022, doi:10.3390/v14102300_

Round 1

Reviewer 1 Report

This manuscript describes the isolation and characterization (phenotypically and genotypically) of a novel temperate phage infecting Bacillus. This phage, called B13, was isolated using Mitomycin C induction from the genome of B. cereus VKM B-13. B13 shares only 31% of its genome with its closest relative, and thus it belongs to a new genus. This manuscript is clearly written, and it was very easy to read. I have only two very minor comments:

- Figure 1: I don’t feel that this is very informative. I think that it is better to just show one of the blowups of the plaques to see the morphology of the plaques. I can’t se a real difference in the current state of the figure between the 24 and the 72 hours, so 1 of them is enough. My guess is that the size of the blowup is 1x1 cm, but a scale (with numbers) should be part of the image. If possible, I think imaging a plate with a lower phage titer will give a clearer plaque phenotype. Additionally, it will be more informative if you’ll compare the phenotype of the plaque on the VKM B-370 strain with an additional strain that showed clear plaques.  

Section 3.4.2: Last sentence – you cannot know what are the characteristics of this gene, so please don’t state that it may be better…

Author Response

Dear Reviewer,

Thank you very much for appreciating our work and for giving us valuable comments.

We agree that the 72-h blowup in Figure 1 was not that informative. Following your recommendation, we have replaced Figure 1, leaving only a 24-h plate with a lower phage titer. We have also added scales with numbers and increased the blowups to 1x1 cm. We have made minor corrections in the paragraph of the manuscript related to this figure.

Generally, the plaque phenotype (clear or turbid) is independent of phage titer in the double agar method. A characteristic feature of temperate phages is the formation of turbid plaques on the lawns of susceptible bacterial strains. Although some temperate phages are able to form clear plaques (due to some peculiarities of the phage-host interaction and/or host metabolism), this phenomenon is more of an exception than the rule. On all the tested strains from our collection, B13 produced turbid plaques similar to those on the lawn of VKM B-370 (Figure 1), therefore, the plaque phenotype comparison you proposed cannot be conducted.

We agree that the last sentence in Section 3.4.2 was overly speculative: “It may be that the B13 endolysin has even better characteristics than Ply57, so an experimental analysis is needed to assess its the stability and activity, which may be the course of our future research.” Thank you for pointing that out. We have replaced the sentence with the following: “Experimental investigation of the B13 endolysin, including its comparative analysis with endolysins from other Bacillus phages, such as Ply57, is of considerable interest and may be one of the directions of future research in our laboratory.”

Reviewer 2 Report

The manuscript by Kazantseva et al describes isolation of the temperate Bacillus phage B13 which has lytic ability on three members of the B. cereus group. After routine thermal and pH tests, the authors sequence and assemble the B13 phage genome. The authors proceed to perform in-depth characterisation of the phage modules (and DNA packaging strategy) prior to performing comparative genomics to show B13 phage remains quite novel in the current landscape, the authors suggesting its identification into a novel genus termed Buntarivirus. I would like to congratulate the authors on such a well organised and written manuscript. It appears to be the exception not the norm these days for manuscripts describing phage genomes to be put together with as much care and attention as this one, as well as providing the reviewers (and the eventual readers) with the required supplementary information. Given this praise, I only have a few (very) minor comments for the authors:

I commend your gene-calling and annotation, it is exceptionally well done. There is just one question mark I have when analysed the genome sequence from Genbank relating to gene with locus_tag: phageB13_37 which runs from 28,420-28,635. There is an overlapping alternate ORF a bit downstream which runs from 28,568-28,906 that may be a better choice for gene-calling in this region. While I appreciate the current annotated ORF (B13_37) would continue an operon, the RBS seems a be a bit further upstream from the proposed start codon than may be expected. The alternate ORF I suggest here has an RBS that is closer to the proposed start codon. That being said, BLASTing either of these CDS’s against Genbank shows both have high coverage (100%)  high identity (>98%) matches to B. cereus group genomes, so it is very difficult to tell which one is the real gene. I ask the authors to at least revisit the analysis on these two ORF’s and I trust they will make an informed choice. 

The cos overhang sequence can be added to the Genbank accession as a misc_feature.

Author Response

Dear Reviewer,

We are eternally grateful to you for appreciating our work and for giving valuable comments, which have helped us in improving the manuscript.

Regarding the ORF with locus_tag: phageB13_37 and the alternative ORF: as you noted, it is very difficult to tell which one is the real gene. We understand your point about that another RBS being closer to the proposed start codon, however, we have decided to leave the start and stop codons of phageB13_37 as predicted by RAST. We feel like we are choosing between two virtually equally probable options, and more compelling evidence is needed for us to discard the automatic annotation.

Thank you very much for the suggestion to add the cos overhang sequence to the Genbank accession as a misc_feature. We have resubmitted the B13 genome with this feature added. The new GenBank version of the B13 genome was uploaded in accordance with the rules on how to upload phage sequences with different types of genome termini, proposed by Russell D.A. [Russell D. A. Sequencing, assembling, and finishing complete bacteriophage genomes //Bacteriophages. – Humana Press, New York, NY, 2018. – С. 109-125.]. For phages using the 3'-cos packaging mechanism, the rule is: “If a 3’cohesive overhang is present it should be at the right end of the genome”.

We regret not uploading the genome properly in the first place, and hope you approve of our decision to adhere to convention. So, the genome has been redeposited and we are currently waiting for the Genbank team to update the accession number and the protein_id numbers. According to the Genbank guidelines, only the version number will be changed in both the accession number (from OP066531.1 to OP066531.2) and the protein_id numbers (for example, from UUW40189.1 to UUW40189.2). In the Manuscript, we have corrected the Accession number and the protein_id numbers accordingly, however, we would prefer to wait until the new version of the genome appears in Genbank so as to make sure all the numbers are accurate, and only then proceed with the publication process. The accession number and the protein_id numbers that need to be clarified before publishing the article are highlighted in the revised version of the manuscript. Also, we had to change the positions of start and stop codons in “Table S2. Annotation of Bacillus phage B13” in Supplementary Information, due to the abovementioned genome reorganization.